# Venous Thromboembolism in Lymphoma: Risk Stratification and Antithrombotic Prophylaxis

**DOI:** 10.3390/cancers12051291

**Published:** 2020-05-20

**Authors:** Stefan Hohaus, Francesca Bartolomei, Annarosa Cuccaro, Elena Maiolo, Eleonora Alma, Francesco D’Alò, Silvia Bellesi, Elena Rossi, Valerio De Stefano

**Affiliations:** 1Dipartimento di Scienze Radiologiche ed Ematologiche, Università Cattolica del Sacro Cuore, 00168 Rome, Italy; francesco.dalo@unicatt.it (F.D.); elena.rossi@unicatt.it (E.R.); valerio.destefano@unicatt.it (V.D.S.); 2Dipartimento di Diagnostica per Immagini, Radioterapia Oncologica ed Ematologia, Fondazione Policlinico Universitario A. Gemelli, IRCCS, L.go A. Gemelli, 8, 00168 Rome, Italy; francescabarto@hotmail.it (F.B.); annarosa.cuccaro@gmail.com (A.C.); elenam86@hotmail.it (E.M.); eleonora.alma@gmail.com (E.A.); silvia.bellesi@policlinicogemelli.it (S.B.)

**Keywords:** venous thromboembolism, lymphoma, Non-Hodgkin lymphoma, Hodgkin lymphoma, risk factors, prophylaxis

## Abstract

Lymphoma is listed among the neoplasias with a high risk of venous thromboembolism (VTE). Risk factors for VTE appear to differ from risk factors in solid tumors. We review the literature of the last 20 years for reports identifying these risk factors in cohorts consisting exclusively of lymphoma patients. We selected 25 publications. The most frequent studies were analyses of retrospective single-center cohorts. We also included two reports of pooled analyses of clinical trials, two meta-analyses, two analyses of patient registries, and three analyses of population-based databases. The VTE risk is the highest upfront during the first two months after lymphoma diagnosis and decreases over time. This upfront risk may be related to tumor burden and the start of chemotherapy as contributing factors. Factors consistently reported as VTE risk factors are aggressive histology, a performance status ECOG ≥ 2 leading to increased immobility, more extensive disease, and localization to particular sites, such as central nervous system (CNS) and mediastinal mass. Association between laboratory values that are part of risk assessment models in solid tumors and VTE risk in lymphomas are very inconsistent. Recently, VTE risk scores for lymphoma were developed that need further validation, before they can be used for risk stratification and primary prophylaxis. Knowledge of VTE risk factors in lymphomas may help in the evaluation of the individual risk-benefit ratio of prophylaxis and help to design prospective studies on primary prophylaxis in lymphoma.

## 1. Introduction

Venous thromboembolism is associated with increased morbidity and mortality among patients with neoplastic diseases [1]. Diagnosis and management of thrombotic events interrupt essential anti-neoplastic treatment. VTE occurring during anti-neoplastic treatment represents a preventable complication causing a high economic burden [2]. Lymphomas are among the malignant diseases at high risk for VTE [3]. Routine assessment of VTE risk is recommended for all patients with newly-diagnosed neoplastic diseases, using validated VTE risk models [4,5]. Khorana et al. developed a risk model for predicting chemotherapy-associated VTE based on baseline clinical and laboratory variables [3]. Only a minority of patients (12.6%) in this study had lymphoma. Several studies indicate that risk factors for VTE in patients with lymphoma are different from risk factors in patients with solid tumors [6,7,8,9]. To provide more information on the VTE risk in patients with lymphoma, we conducted a systematic review of the literature to determine the incidence of VTE in patients with lymphoproliferative disease and to identify disease and patient characteristics associated with the greatest risk for VTE. VTE risk factors in lymphoma differ from VTE risk factors in solid tumors that have been used to build pan-cancer VTE risk scores, which do not capture the disease-specific VTE risk in lymphomas. As physicians increasingly specialize in the treatment of a few or single cancer type, knowledge of disease-specific risk factors will become more and more important to help treating physicians with their decisions on VTE prophylaxis. More research is needed to assess, validate, and improve VTE risk scores in patients with lymphoma.

## 2. Methodology

To review VTE risk factors in patients with lymphoma, we screened the Pubmed database for reports published between 1 January 2000 to 31 December 2019, using the MeSH terms “lymphoma” and “thromboembolism” and “venous”. We reviewed 246 references. Publications addressing VTE risk in cohorts of adult patients with lymphoma were included for this review, and 21 studies were eligible (Figure 1).

Four additional studies were identified by cross-referencing in the 21 published reports. The articles were divided among all the authors for a first classification and summary and then reviewed by the two senior authors (S.H. and V.D.S.). Of the 25 studies included, 12 studies were retrospective single-center studies [7,8,10,11,12,13,14,15,16,17,18,19], two studies were retrospective multicenter trials [20,21], two pooled analysis of clinical trials [9,22], two meta-analyses [23,24], two analyses of patients [25,26] and three analyses of population-based registries [27,28,29], and two prospective single-center studies [6,30] (Table 1).

Patients included into these studies were identified by institutional databases in 14 studies [7,8,10,11,12,13,14,15,16,18,19,20,21,30], by the local hospital discharge registry in one study [17], by cancer registries in four studies (California Cancer Registry [27], VA Cancer registry [25], Swedish cancer registry [29], Danish lymphoma database [28]), by clinical trial registries in five studies [6,9,22,24,26], and by published patient data in one study [23]. The composition of patient cohorts varied widely both in terms of numbers, ranging from 42 to 16,755 patients, and in types of histologies included (Table 1). Ten studies analyzed the VTE risk in patient cohorts with a single histology [6,9,10,16,17,18,19,20,21,29]. Other cohort restrictions were, for example, the use of almost only male patients in the VA Cancer registry [25], only patients receiving at least one chemotherapy cycle [15], patients who had at least one hospital admission [8], or patients who had at least one serum sample available [6]. The method to identify VTE differed: record review was the most frequent tool for event identification. ICD codes were used to identify VTE in patient registries [17,25,27,28]. Other methods used included pharmacovigilance reports in a clinical trial [22] and a survey of participants of long-term survivors after bone marrow transplantation with a questionnaire [26]. The validity of coded discharge diagnoses for VTE identification was assessed in a study from Denmark [17]. The positive predictive value of the VTE discharge diagnosis was 85%, while the sensitivity of the VTE discharge diagnosis was only 53% [17].

Borg et al. note that patients may have long admissions, and therefore, an episode of VTE may get lost in other problems, and therefore not be registered [17]. Another cause of underreporting could be the exclusion of events, for example, upper extremity thrombosis, either for diagnostic uncertainty or as the presence of central venous catheters (CVC) could be the major contributing factor [27,28].

## 3. Epidemiology

Caruso et al. have analyzed the risk of VTE in a meta-analysis of 18 published studies, including 18,018 patients, the largest lymphoma population that has been analyzed and published so far [23]. The incidence rate (IR) of VTE for patients with NHL was 6.5% (95% CI, 6.1–6.9%), significantly greater than that observed for HL patients with an IR of 4.7% (95% CI, 3.9–5.6%). In patients with aggressive lymphomas, the IR of events increased to 8.3% (95% CI, 7.0–9.9%) [23]. Data on the second-largest population of lymphoma patients analyzed for VTE risk was published by Mahajan [27]. VTE was diagnosed within two years in 670 patients (4.0%). The rate of development of VTE was highest in the first year (47 events/1000 patient-years) and fell sharply over time (7 events/1000 patient-years in the second follow-up year) [27]. These IR are similar to another population-based study from Denmark (Lund) [28] or from pooled analyses of clinical trials [9,22]. IR reported from single-center studies tended to be higher, ranging from 10–15% (Table 1) [6,8,12,13,15,16,17,19,20,21]. Some of these studies included only patients with aggressive histology, like diffuse large B cell lymphomas (DLBCL) [6,16,17,19,20]. The highest IR ever reported for VTE in patients with lymphoma was 59.5% in patients with primary CNS lymphoma (PCNSL) [18]. Differences in reporting could be one possible explanation for these differences. Heterogeneity in risk factors for VTE present in the study population may be another reason for these differences. Risk factors for VTE in lymphoma will now be explored in the next paragraphs.

## 4. Risk-Modifying Factors

In a simplified model, risk factors could be classified as modifying the VTE risk by their interplay with factors pertaining to Virchow’s triad [31,32,33]: hemostasis results from bed rest and vascular compression by the tumor mass; vessel injury is caused by intravasation of cancer cells, intravascular devices, and systemic therapies; and hypercoagulability results from the production of procoagulants and interaction of neoplastic cells with host cells, including platelets and leukocytes, mediated by direct cellular interactions and inflammatory cytokines. We will now review clinical and laboratory risk factors that are either related to the patient or the disease (Table 2). Thromboprophylaxis with low molecular heparin (LMWH) might modify VTE risk. However, the studies on VTE risk factors do not report sufficient data to evaluate the role of thromboprophylaxis.

### 4.1. Individual Patient-Related Factors

#### 4.1.1. Age

Older age has been reported as a risk factor for VTE in several studies on cancer-associated thrombosis. Six retrospective studies (four single-center, two multicenter) from Asia and Italy found an increased risk for VTE in patients with NHL older than 60 or 65 years [6,8,19,20,22,30]. The majority of the patients included in these studies had diffuse large B cell lymphoma. Restricting the analysis to DLBCL, Park et al. confirmed the contribution of age older than 60 years to the VTE risk [30]. The odds ratios for older age in the five studies vary between 1.6 and 3.3. One population-based study from California analyzing data of 16,755 patients with NHL identified an age of 45 years and above as risk factor for VTE (HR, 1.4, 95% CI, 1.1–1.7) [27]. Data from other studies did not point to an association between age and risk of VTE in lymphomas (Table 2).

#### 4.1.2. Gender

Studies on the association between gender and risk of VTE in lymphomas point to the female gender as a potential risk factor. However, data are far from being conclusive. Evidence for an association between female gender and VTE comes from three studies [15,20,22]. An increased risk for VTE was observed in a multicenter study on 235 Asian patients with PCNSL (HR 2.3; 95%; CI 1.1–5.0) [20]. Female gender was a risk factor for VTE in multivariate analysis of 422 lymphoma patients treated at the MD Anderson Cancer Center (OR 3.51; 95%, 1.67–7.40) [15]. The pooled analysis of 12 clinical trials from Italy identified female gender as a risk factor for severe VTE grade 3 or more [22]. However, the female gender was not associated with risk for VTE in 14 other studies (Table 2).

#### 4.1.3. Obesity

Obesity is well known as being a VTE risk factor in the general population and is a variable in the Khorana CAT assessment score [3]. Data on an association between body mass index (BMI) and VTE risk in patients with lymphoma are inconsistent. Increased BMI was identified as a VTE risk factor in three studies, while six studies did not find an association (Table 2). Antic et al. found a BMI > 30 kg/m^2^, present in only 1.5% of patients, as a strong risk factor both in univariate and multivariate analysis of 1820 patients with lymphomas (OR 10.7; 95% CI, 3.3–34.6), including this parameter as a variable in the ThroLy score [7]. In the analysis of 2730 male NHL patients of the VA Cancer Registry in the US, BMI > 30 kg/m^2^ was present in 25% of patients and a moderate VTE risk factor in a competing risk model for VTE in the first year of NHL diagnosis (adjusted HR 1.6; 95% CI 1.08–2.37) [25]. A BMI > 25 kg/m^2^ was identified as a VTE risk factor in long-term survivors following allogeneic transplantation for NHL [26]. However, BMI was not included in the analysis of VTE risk in the majority of published studies on lymphoma patients (Table 2).

#### 4.1.4. Performance Status/Immobility

Immobility due to poor performance status, hospitalization, or post-surgery is one of the most important risk factors for VTE for patients with lymphoma as for other cancers. Khorana reported VTE events in 4.1% during hospitalization of cancer patients, which is clearly higher than the 1.6% incidence in the study cohort of cancer outpatients used for the development of the Khorana score [3,34]. Only one out of ten studies that analyzed performance status as a VTE risk factor in patients with lymphoma did not find an association between reduced performance status and VTE risk (Table 2) [16]. Most studies defined a reduction of the daily activity of the patient with variation from partial to complete immobilization corresponding to ECOG scale grade 2 or more as poor performance. This threshold is not only a VTE risk factor, but also a poor prognostic parameter for lymphoma-specific and overall survival in nearly all types of lymphoma, as it is one of the variables in the International Prognostic Index (IPI) [35]. The prevalence of reduced performance status in the study cohorts varied widely depending on the clinical setting and lymphoma type. The frequency of patients with a reduced performance status was lowest in ambulatory cohorts [7,12] and highest in a patient cohort with PCNSL (50%) [20]. In a population-based database from Denmark, the proportion of patients with reduced performance status was 17% [28]. The OR of reduced performance status for VTE ranged between 1.5 to 5.1 [6,7,8,12,17,20,28,30], with one outlier of 39.9 [19].

A lower threshold to define poor performance status as a VTE risk factor was found in the analysis of the Danish lymphoma database of 10,375 patients, setting the cut-point at ECOG grade 1 [28]. This corresponds to symptoms that do not interfere with daily activity and do not lead to increased immobility. In our single-center study, poor performance status was associated with a higher VTE localization to the lower extremities [8].

#### 4.1.5. Comorbidity

Comorbidities such as congestive heart failure, renal, liver, and pulmonary disease have been reported as VTE risk factors in hospitalized cancer patients [36]. The impact of comorbidities on the VTE risk has been only rarely addressed in patients with lymphoma. Mahajan et al. used the California Cancer Registry coupled with the California Patient Discharge database to determine the incidence of first-time VTE in 16,755 patients with lymphoma and found that a greater number of chronic comorbid comorbidities was a strong predictor for VTE [27]. Patients with one or two comorbidities in addition to lymphoma were 2-fold and patients with three or more comorbidities were 4-fold more likely to develop VTE. The types of comorbid conditions were not reported. In the study by Park et al. on 686 patients, hypertension was significantly associated with VTE (*p* = 0.017), but was not maintained as a significant parameter in the competing risk analysis [30]. The diagnosis of coronary artery disease was identified as a risk factor for late VTE in long-term survivors with lymphoma following autologous transplantation [26].

#### 4.1.6. Prior Thrombosis/Thrombophilia

A previous thrombosis has been reported in some studies to be associated with an increased risk for VTE after diagnosis of lymphoma and during treatment [7,12,17,25]. Data are not conclusive, as many studies did not register prior VTE as a variable, or the study size was too small, and no prior VTE in the study cohort had been observed [10,18,19,20]. Prior VTE is a relatively rare event. In the cohort of 2730 NHL patients of the Veteran’s Administration Central Cancer Registry, a previous VTE was recorded in 1.6% of patients [25]. Data interpretation is further complicated by differences in the type of thrombotic event counted as prior thrombosis, as well as the lack of information between the time between the prior VTE and lymphoma diagnosis/treatment. Antic et al. combined prior VTE, myocardial infarction (MI), and stroke as a single variable to develop the ThroLy score [7]. This variable that was present in 1% of patients had the highest OR (14.1; 5% CI, 4.4–45) in the multivariate analysis and was assigned 2 points in the 7-parameter, 10-point predictive model for VTE, the ThroLy score. In a validation study of the ThroLy score in a single-center cohort of 428 patients, Rupa-Matysek et al. confirmed the presence of previous VTE/MI/stroke as a significant predictive parameter [12]. Sanfilippo et al. counted only VTE as prior thrombosis and found an adjusted hazard ratio of 4.73 (95% CI, 2.47–9.04) for a history of VTE [25]. One cannot exclude that some of the prior VTE could be already heralding events of the lymphoma activity [37]. In the analysis of our cohort of 857 patients with lymphoma, we counted 54 VTE that were present at diagnosis or occurred in the 6 months prior to the diagnosis as heralding event [8].

A genetic component appears to play a role in cancer-associated thrombosis [38,39]. The contribution of hereditary or acquired thrombophilia as a VTE risk factor in patients with lymphoma is yet to be explored in comprehensive studies. In a study on 70 patients with splenic marginal zone lymphoma, Gebhart et al. observed a high prevalence of anti-phospholipid antibodies [21]. Lupus anticoagulans activity was present in 9/70 (13%) patients and was associated with a higher VTE risk, in particular following splenectomy. In a cohort of 142 Japanese patients with DLBCL, the investigators screened for the possibility of inherited thrombophilia by measuring antithrombin activity, protein C activity, and protein S antigen only in the 15 patients that developed VTE [19]. None of them had inherited thrombophilia. We currently address the contribution of hereditary and acquired thrombophilia in an ongoing prospective study on the risk of VTE in patients with lymphoma (VANILLA study).

### 4.2. Lymphoma-Related Factors

#### 4.2.1. Histology

Lymphoma histologies can be roughly divided into indolent and aggressive lymphomas, with the latter category including intermediate and highly-aggressive lymphoma types. The literature consistently reports that aggressive histology in NHL associates with increased VTE risk [7,8,12,13,14,22,23,25,27,28,30]. Aggressive histology has been identified in single-center, multi-center, and population-based cohorts. In the group of B cell-NHL, DLBCL is the most frequent subtype and is clearly at a higher VTE risk with respect to follicular and other indolent lymphoma [8,12,14,22,27]. Mantle cell lymphoma is considered an aggressive B cell lymphoma. Peripheral T cell lymphomas generally have a poor prognosis and are included in the group of lymphoma types with a higher risk for VTE [8,28]. Ten studies presented in Table 1 focus on one lymphoma type, most DLBCL. The actuarial incidence rate in the first year after diagnosis for DLBCL is about 10–12%. The incidence of VTE for localization of subtypes of aggressive B cell lymphoma to particular sites, as CNS and mediastinum are discussed in the next paragraph. In indolent lymphomas, the incidence rate varies between 1.5 and 4% [25,28]. The risk for VTE in the first year after diagnosis has been reported to be 4-fold higher in 2190 patients with Waldenstroem’s macroglobulinemia/lymphoplasmocytic lymphoma with respect to a control population, identified through Swedish registries [29]. The total incidence of VTE in patients with Hodgkin lymphoma treated in the GHSG is 3.3% [9].

#### 4.2.2. Site of Disease

Lymphoma localization to the CNS has the highest VTE risk ever reported in patients with lymphoma. In a series of 42 patients with primary CNS lymphoma (PCNSL), Goldschmidt reported VTE in 24 patients (59.5%) [18]. In a multicenter study from Korea including 235 patients with PCNSL, 33 patients (14%) developed VTE during the 21 months follow-up period [20]. In our single-center study including 857 patients with lymphoma, we found that the 33 patients with PCNSL had a peak VTE incidence of 27.2% [8]. In a single-center study from Korea, brain involvement in 51 of 686 patients was associated with 19.6% incidence rate of VTE, translating into a 2.04-fold (95% CI, 1.03–6.30) increased risk [30]. In a Danish population-based study CNS involvement that was present in 287 of 10,375 patients was associated with a cumulative 2-year incidence of VTE of only 8.4% increasing the VTE risk by 2.52 fold (95% CI, 1.54–4.12) compared to patients without CNS involvement [28]. This study reported a lower VTE incidence among all lymphoma patients compared with prior studies.

Mediastinal localization as a bulky disease is another site of particular risk for VTE. In a series of 42 patients with primary mediastinal B cell lymphoma, 15 patients (35.7%) developed VTE [10]. In 10/15 patients the thrombosis was present at diagnosis. Antic et al. identified mediastinal and extranodal localization as VTE risk factors [7]. In particular, mediastinal involvement was associated with an 8-fold increased VTE risk, while extranodal localization increased the risk for VTE by the factor 2.3. In the ThroLy score Antic et al. assigned two points to mediastinal involvement and one point to extranodal localization [7]. Both factors were confirmed in the studies by Rupa-Matysek [12,13]. We found bulky disease defined as 10 cm independent of mediastinal localization as a risk factor both in univariate and multivariate analysis with an OR of 3.23 (95% CI, 1.85–5.63) [8].

#### 4.2.3. Stage of the Disease

A higher tumor burden results in a higher risk for VTE, and VTE, in turn, is a marker of tumor aggressiveness and poor prognosis [27]. Parameters for tumor burden are the stage of disease according to the Ann Arbor staging system, with stage I/II considered localized disease, stage III with disease extension on both sides of the diaphragm, and stage IV with disease spreading to extranodal organs as the advanced stage. The advanced-stage disease has been associated with the VTE risk in large cohort studies on B-NHL (HR 1.49; 95% CI 1.0–2.00 [25]; and 1.5; 95% CI, 1.2–1.7 [27]) and in DLBCL case series (3.31, 95% CI, 1.55–7.09 [6]; and 2.8, 95% CI, 1.1–6.8 [17]). Six studies did not find a significant association between stage and VTE risk [14,15,16,19,22,23].

#### 4.2.4. Laboratory Variables

Laboratory variables included in the Khorana risk assessment model for VTE in cancer are a pre-chemotherapy platelet count of 350 × 10^9^/L or more, hemoglobin level less than 100 g/L, and pre-chemotherapy WBC count > 11 × 10^9^/L [3]. There is no study that indicates that the platelet count is a VTE risk factor in lymphomas (11 negative studies) (Table 2). A WBC count > 11 × 10^9^/L has been identified as VTE risk factor only in two studies on lymphomas [6,12], while nine other studies were negative [7,8,9,10,15,17,19,20,28]. In the same line, data on anemia are inconsistent; seven studies did not find an association of pre-chemotherapy hemoglobin levels with VTE risk [6,8,9,12,17,19,28], one study found an association only in univariate analysis [25], and the only study that identified Hb < 100 g/L as VTE risk factor also in multivariate analysis was the study by Antic et al. [7]

Alterations of other laboratory values have been occasionally been associated with an increased VTE risk in lymphoma. Elevated levels of LDH indicating a more aggressive disease were predictive of VTE in some studies [8,28,30,40], but not in others [6,15,16,17,20,25]. Low albumin levels that were associated to VTE risk in the study form our institute [8] and a study on PCNSL from Korea [20] might be indicators of inflammatory status, as albumin inversely correlates with IL-6 levels [41]. Higher IL-6, IL-10, RANTES, and IP-10 levels, but not TNF alpha, were associated with increased VTE risk in a cohort of 322 patients with DLBCL form Korea; however, no parameter retained statistical significance at the multivariable analysis [6]. Another emerging parameter is the mean platelet volume (MPV) [42,43]. A lower MPV has been associated with the risk of VTE in cohorts of patients with DLBCL and HL, however, with varying cut-points [44,45].

#### 4.2.5. International Prognostic Index (IPI)

The International Prognostic Index (IPI) has been for more than 25 years the most widely-used prognostic score to predict prognosis in aggressive lymphomas [35]. As all of the factors that compose the IPI have been—albeit variably—associated with an increased VTE risk in lymphomas, such as age, stage, performance status, LDH levels and the number of extranodal sites, it appears consequential that the IPI score has also been found to be associated with VTE risk in several studies [6,13,16,17,19].

#### 4.2.6. Time after Diagnosis

The VTE risk in lymphoma patients is the highest upfront during the first months after lymphoma diagnosis and thereafter decreases over time. This upfront risk may be related to the tumor burden and the start of chemotherapy as contributing factors. Zhou et al. reported that 64% (51/80) of the patients with venous thromboembolism experienced the event before or during the first three cycles of therapy [15]. The most frequently-reported median time to VTE was about 2 months [6,17,20,22,30]. For patients with DLBCL, Sanfilippo et al. reported a VTE incidence rate of 10% during the 6 months treatment period and a drop to 2% in the post-treatment period from 6 months to 2 years [25]. Caruso et al. concluded from a meta-analysis of 18 studies that 95% of VTE occurred during the treatment period and only 1.2% in the follow-up period [23].

The presence of a VTE prior to the start of chemotherapy is particularly frequent in patients with local venous compression. Lekovic et al. found a high incidence rate in PBMCL with mediastinal masses (35.7%), and 10/15 (66.7%) VTE were already present at diagnosis [10]. The proportion of VTE occurring before the start of therapy of all VTE that had been observed varied between 16/80 (20%) 10/27 (37%) and 54/95 (57%) in three retrospective single-center studies [8,14,16]. These figures are much lower in pooled analyses of study protocols and meta-analyses. Caruso et al. calculated a 3.8% proportion of VTE at disease presentation in the meta-analysis of 18 studies [23]. The analysis of three protocols of the German Hodgkin Study group revealed only a few events between diagnosis and the start of chemotherapy (5/175, 2.9%) [9]. In the pooled analysis of 12 study protocols, only VTE occurrences after study enrolment were registered [22]. Differences in cohort compositions and in reporting VTE events before the start of therapy may contribute to the heterogeneity of the results.

#### 4.2.7. Type of Therapeutic Regimen

The contribution of single chemotherapeutic agents to the VTE risk is often difficult to isolate from other contributing risk factors, as therapeutic regimens may vary according to the aggressiveness of the disease. Patients with advanced stage Hodgkin lymphoma treated with BEACOPP instead of ABVD have a higher VTE risk [9]. The randomized comparison between BEACOPP schedules administered every 14 days instead of 21 days was associated with a higher VTE incidence rate [9]. Sanfilippo et al. found that the addition of doxorubicin to the CVP regimen increased the VTE risk in patients with B cell lymphomas (DLBCL and FL) [25]. This difference remained significant even when adjusting for histology, as DLBCL patients were much more likely to receive doxorubicin. Regimens that contain methotrexate and/or doxorubicin such as hyper-CVAD, CHOP, and ABVD been described to be associated with an increased VTE risk in a single-center study from the MD Anderson Cancer Center of 422 patients with a variety of lymphomas when compared to regimens not including these two drugs [15]. Lenalidomide is associated with an increased risk for VTE in myeloma. A recent meta-analysis showed that the rate of VTE in B-cell NHL patients treated with lenalidomide in clinical trials is similar to the rate in multiple myeloma [24]. The VTE rate appears to be lowest for lenalidomide combined with a biologic (0.49/100 patient-cycles) compared with single-agent lenalidomide (1.07) or its combination with chemotherapy (0.89) [24].

#### 4.2.8. Indwelling Central Venous Catheters

Indwelling central venous catheters (CVC) are associated with increased risk of VTE also in patients with lymphoma. There are only a few studies systematically addressing the VTE risk of CVC [11,28,30]. Some studies report CVC-related VTE without formally analyzing CVC as a risk factor [14,19,28], and one study excluded CVC-related upper extremity DVT [27]. In a single-center study from China, the incidence of upper-extremity deep vein thrombosis related to the presence of a peripherally-inserted CVC (PICC) was 7.1% (40/565) in patients with lymphoma, without significant differences between lymphomas, but significantly higher than in patients with other cancers (209/7463, 2.80%) [11]. In a study from Korea on 686 patients with lymphoma, the incidence of VTE in patients with a CVC (42/460, 9.1%) was higher than in patients without a catheter (12/226, 5.3%, *p* = 0.042) [30]. However, only three of the 42 cases were catheter-related venous thrombosis, suggesting that in most patients, VTE was not directly associated with the presence of CVC, but was due to other contributing factors such as chemotherapy. In a study linking the Danish lymphoma database with the Danish patient registry, Lund found that in 1453 of 10,375 (14%) patients CVC were used, and the use of a CVC was associated with a 6.67-fold increase in the risk of VTE (95% CI: 1.18) [28]. However, coding in the registry did not allow for reliable identification of upper-extremity VTE in order to establish a link between CVC and CVC-related VTE.

#### 4.2.9. Supportive Care Agents

Cytopenias and the administration of erythropoiesis stimulating agents (ESA) and myeloid growth factors, such as granulocyte colony stimulating factor (G-CSF), have been identified as risk factors for cancer-associated thromboses [3]. Supports with ESA were associated with an increased risk of VTE and mortality in patients with cancer [3]. This has not been specifically addressed in studies on VTE risk in lymphoma patients with lymphoma. Anemia and neutropenia during chemotherapy have been associated with the VTE risk in the study by Anti et al., and could be indicators of the use of growth factors, but formal proof is missing in lymphoma patients [7].

## 5. Risk Assessment Models

International guidelines recommend risk factor assessment in patients with cancer [4,5]. The most widely-used model was developed by Khorana et al. for patients receiving chemotherapy in an outpatient setting [3]. Five studies assessed this model in a cohort of exclusively lymphoma patients [6,8,9,13,22]. Santi et al. found that a Khorana score of ≥3 was associated with the incidence of VTE in a pooled analysis of 12 lymphoma studies [22]. Four other studies failed to find evidence for an association [6,9,13]. In the same line, analyses of patient cohorts focusing on specific cancer sites, such as pancreatic cancer or lung cancer, showed poor performance of the Khorana score [46,47]. A recent meta-analysis revealed that the predictive power of the Khorana score was not homogenous across various types of cancer [48]. The pan-cancer Khorana score does not capture the disease-specific characteristics associated with VTE risk, and clinicians should be cautious when applying the Khorana score as a universal risk assessment tool.

Moreover, the sensitivity of the Khorana score is quite poor. Most VTE events in cancer patients occur outside the high-risk group [49]. In a meta-analysis on 27,849 patients, only 23.4% of VTE occurred in the 17.4% of patients with a high-risk (>3) Khorana score [49]. When decreasing the threshold to include also patients with a risk score of 2, the detection rate increased to 55.2% of VTE events in a patient cohort that included 47% of the cancer population [49].

Variations of the Khorana risk score have been developed to improve risk assessment that include additional parameters, such as metastatic disease, vascular compression, and previous VTE, such as in the ONKOTEV score [50] or combining clinical and genetic risk factors in the TiC Onco score [51]. Pabinger et al. identified the tumor-site risk category as the only parameter of the Khorana score to predict VTE in the Cancer and Thrombosis Study (CATS) cohort of 1423 patients [52]. Combining tumor-site category and D-dimer values, the authors developed a simple and VTE risk model validated in an independent patient cohort. The CATS cohort included 249 patients with lymphoma (17%), while the validation cohort of 832 patients did not contain lymphoma patients. This VTE risk score merits further exploration in lymphoma patients.

Recently, Antic et al. developed a lymphoma-specific, 7-parameter 10-point score they termed ThroLy score [7] that was validated so far in one single-center study [12]. Similar to the Khorana score, the performance of the ThroLy score is limited by a high frequency of VTE occurring in the low-risk group. In a validation study by Rupa-Matysek, 48% of VTE occurred in the low-risk group of the ThroLy score that comprised 75% of patients [12].

We developed a simple score based on only 3 parameters, CNS involvement, bulky disease and performance status, with CNS involvement defining the highest risk group, and patients with ether bulky disease and/or reduced performance status as high-risk patients, and all other patients as standard-risk patients [8]. The predictive performance of our score is quite good when compared to the Khorana and ThroLy scores. Our VTE score identified 82% of VTE in the high-risk group that consisted of 48% of patients [8]. If the performance will be confirmed in validation studies, this score would provide a simple tool for VTE risk stratification and could be useful in designing studies on primary prophylaxis for higher-risk patients.

Further studies are clearly needed to develop robust lymphoma-specific VTE scores. As VTE risk may change during therapy due to periods of immobilization, positioning of CVC, or changes in therapeutic regimens, the development of dynamic risk models might be more helpful to guide VTE prophylaxis.

## 6. Treatment and Prophylaxis of VTE in Lymphomas

There are no lymphoma-specific guidelines for treatment or prophylaxis of VTE. VTE should be treated in accordance with international guidelines for patients with cancer, such as those endorsed by the International Society on Thrombosis and Haemostasis [4] or the American Society of Oncology [5]. In brief, these guidelines recommend low-molecular-weight heparin (LMWH) for the initial treatment of established VTE in patients with cancer when creatinine clearance is ≥30 mL per min. [4]. After 6 months, termination or continuation of anticoagulation should be based on an individual evaluation of the benefit–risk ratio, tolerability, drug availability, patient preference, and cancer activity. For the treatment of symptomatic catheter-related thrombosis, the international guidelines recommend anticoagulant treatment for a minimum of 3 months, and as long as the CVC is in place [4].

Thrombocytopenia can develop during chemotherapy and treatment of VTE. An expert panel endorsed by the Gruppo Italiano Malattie Ematologiche dell’Adulto Working Party on Thrombosis and Haemostasis produced a formal consensus about platelet cut-offs for safe treatment with LMWH in patients with hematological neoplasms and thrombocytopenia [53]. Dose modifications of LMWH for platelet counts < 50 × 10^9^/L are recommended. In clinical practice, risk factors for bleeding and VTE have to be considered to balance the risk in the individual patient, as localization to particular sites could potentially increase the bleeding risk (e.g., localization of the lymphoma in the gastrointestinal or the central nervous system).

The use of direct oral anticoagulants (DOAC) is emerging as a safe and effective alternative to subcutaneous LMWH for the treatment of cancer-associated VTE. Four randomized clinical trials showed that DOAC were non-inferior to LMWH for the treatment of cancer-associated VTE without an increased risk of major bleeding [54,55,56,57]. Only a few patients with hematological malignancies were enrolled, and the proportion of lymphoma patients was lower than 5%.

In the Hokusai VTE cancer study, 40 of 1050 (3.8%) patients had lymphoma [54]. The primary outcome was the composite of recurrent VTE or major bleeding. The DOAC edoxaban was not inferior to LWMH. In a recent subgroup analysis, data for 111 patients with hematological malignancies including the 40 patients with lymphoma were presented, and no difference in the primary outcome during the 12-month observation period between edoxaban and dalteparin was observed (8.9% and 10.9%) [58].

The SELECT D trial randomized 406 patients with active cancer and VTE to either rivaroxaban or dalteparin [55]. No differences were observed between them in the primary outcome, which was recurrent VTE. Only 10 (2.5%) patients had hematological cancers.

In the ADAM VTE trial, the safety of the DOAC apixaban was compared to LMWH in 300 cancer-associated VTE [56]. The primary outcome was major bleeding, and secondary outcomes included VTE recurrence and a composite of major plus clinically-relevant non-major bleeding (CRNMB). No major bleedings were observed in the apixaban arm, and VTE recurrence was lower in the apixaban arm. Only 16/300 (5.3%) patients had lymphoma.

In the Caravaggio trial, apixaban was compared to dalteparin for the treatment of cancer-associated VTE [57]. Dalteaparin was noninferior to dalteparin without an increased risk of major bleeding. Of the 1155 enrolled patients, 85 (7.3%) had hematological malignancies, and a sub-analysis of the small number of hematological malignancies did not reveal a difference between the two arms.

International guidelines do not recommend routine primary VTE prophylaxis with LMWH in ambulatory patients receiving systemic anti-cancer therapy [4]. Guidelines recommend prophylaxis with LMWH or fondaparinux when creatinine clearance is ≥30 mL/min, or with unfractionated heparin in hospitalized patients with cancer and reduced mobility [4]. Reduced mobility expressed by the ECOG performance status is a risk factor in 9/10 studies addressing this parameter in patients with lymphoma (Table 2). Therefore, hospitalization associated with conditions limiting the patient’s mobility should be a reason for thromboprophylaxis.

The guidelines do recommend primary prophylaxis in ambulatory patients who are receiving systemic anti-cancer therapy at intermediate-to-high risk of VTE, identified by cancer type (i.e., pancreatic) or by a validated risk assessment model (i.e., a Khorana score ≥ 2), and not actively bleeding or not at a high risk of bleeding [4]. As there is no widely-accepted and validated VTE risk assessment model for lymphomas, no general recommendations for primary prophylaxis in ambulatory patients with lymphoma receiving systemic anti-cancer therapy can be given. Data on the use of LMWH in the studies presented in this review are insufficient to draw any conclusion on the role of primary prophylaxis. The decision to start primary prophylaxis in ambulatory patients with lymphoma should be based on individual evaluation of the benefit–risk ratio, taking into consideration the VTE risk factors described in this review, tolerability, patient preference, and risk of bleeding. We consider localization to the CNS, venous compression by a locally-advanced mass, and reduced performance status as the major VTE risk factors, and consider primary prophylaxis in these patients when they are at low risk for bleeding. The role of primary prophylaxis in ambulatory patients with lymphoma has to be addressed in prospective clinical trials.

Recently, DOACs have also been compared to the use of placebo as primary prophylaxis to prevent cancer-associated VTE in high-risk patients, identified by the Khorana score (>2). In the AVERT trial, 574 patients were randomized, and the occurrence of VTE was lower in the apixaban group (4.2% vs.10.2%), while major bleeding was increased (3.5% vs.1.8%) [59]. A total of 145 (25.2%) patients with lymphoma were included. Outcomes according to disease groups were not reported. In the CASSINI trial randomizing 841 patients, rivaroxaban reduced the occurrence of VTE during the treatment period when compared to placebo (2.6% vs. 6.4% ) without increasing the risk of major bleeding [60]. However, there was no difference in the VTE incidence when the analysis was extended to the full observation period of 180 days. Separate outcome for the 59 (7.0%) patients with lymphoma were not reported.

Primary VTE prophylaxis with DOACs is an interesting perspective that has to be further explored in randomized studies specifically designed for patients with lymphoma. Potential interactions with chemotherapeutic drugs involving metabolism via cytochrome P450 system is of concern. A prerequisite for the design of a randomized study is the validation of the recently-published VTE risk models for lymphomas, as the performance of the Khorana score is low in this category of patients.

## 7. Conclusions

Lymphomas are among the neoplasias at high risk for VTE. Aggressive lymphomas have about a 10–15% incidence rate of VTE in the first year. This risk is even higher when the disease is localized in the CNS or causes a mediastinal mass. The risk is the highest upfront, from diagnosis to the first cycles of antineoplastic treatment. Extensive disease activity, immobility, the positioning of CVC, and administration of chemotherapy with anthracyclines all contribute to the VTE risk upfront. Previous VTE is a risk factor, but well-conducted studies exploring genetic background as a contributing factor are missing. Assessment scores for VTE that were developed for patients with solid tumors, such as the Khorana score, do not predict the VTE risk in lymphoma patients. In the absence of a validated risk score, no evidence-based recommendation for VTE prophylaxis in ambulatory patients undergoing anti-neoplastic treatment can be given. Individual evaluation of the risk-benefit ratio considering the risk factors described in this review is the current strategy. Prospective studies on primary prophylaxis specifically designed for patients with lymphoma are warranted. Moreover, studies to gain more insight into pathogenetic factors that induce VTE in lymphomas are needed.

## Figures and Tables

**Figure 1 cancers-12-01291-f001:**
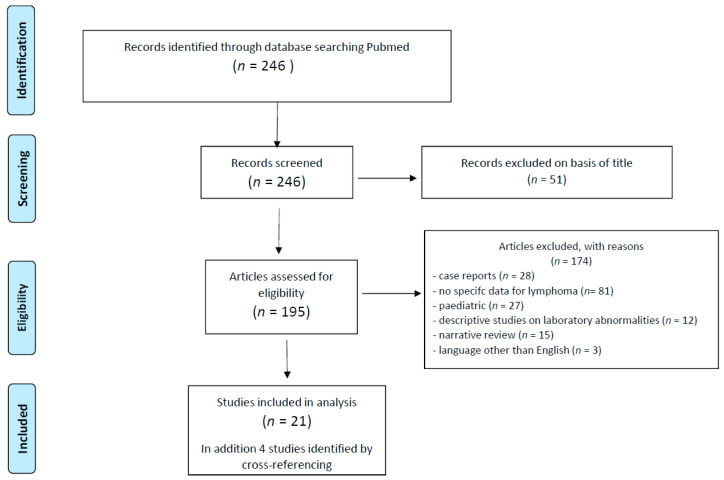
Flow-chart of screening process.

**Table 1 cancers-12-01291-t001:** Characteristics and Incidence Rates of VTE in 25 studies published between 2000 and 2019.

First Author	Year	Country	Ref. No.	Type of Study	No. Pts	Histologies	Median Age	Identification of Events	Median Time	No. VTE	Cumulative Incidence
Sanfilippo	2016	USA	25	registry	2730	DLBCL FL	64	ICD codes	28.4 mo.	246	DLBCL 10% at 6 mo.
Santi	2017	Italy	22	clinical trials	1717	NHL	57	pharmacovigilance	6 mo.	53	2.9% at 6 mo.
Antic	2016	Serbia	7	retrospective single center	1820	NHL, HL, CLL	53	records review	9 mo,	73	5.3% during therapy
Rupa-Matysek	2018	Polonia	12	retrospective single center	428	DLBCL, HL	50	records review	37 mo.	64	15%
Rupa-Matysek	2018	Polonia	13	retrospective single center	428	DLBCL, HL	50	records review	37 mo.	64	15%
Hohaus	2018	Italy	8	retrospective single center	857	NHL, HL	51	records review	15 mo.	95	11.1% at 9 mo.
Park	2012	Korea	30	prospective single center	686	NHL, HL, CLL	51	records review	21.8 mo.	54	7.9% at 1 yr
Mohren	2005	Germany	14	retrospective single center	1038	NHL, HL,	59	records review	n.a.	80	7.7%
Zhou	2010	USA	15	retrospective single center	422	NHL, HL	57	records review	2 yrs	80	17.1% at 2 yrs
Mahajan	2014	USA	27	population-based databases	16755	NHL	n.a.	ICD codes	2 yrs	670	4% at 2 yrs
Lund	2015	Denmark	28	population-based databases	10375	NHL, HL	n.a.	ICD codes	2 yrs	355	3.9%at 2 yrs
Caruso	2010	International	23	meta-analysis	18018	NHL, HL	n.a.	published studies	n.a.	1149	6.4% during therapy
Lim	2016	Korea	6	prospective single center	322	DLBCL	56	not specified	41.9 mo.	34	10.6% at 1 yr
Komrokji	2006	USA	16	retrospective single center	211	DLBCL	57	records review	n.a.	27	12.7% during therapy
Borg	2016	Denmark	6	retrospective single center	289	DLBCL	67	ICD codes	16 mo.	32	11.1% at 2 yrs
Yokoyama	2012	Japan	19	retrospective single center	142	DLBCL	63	records review	n.a.	15	11% during therapy
Goldschmidt	2003	Israel	18	retrospective single center	42	PCNSL	61	records review	n.a.	25	59.5% at 3 mo.
Byun	2019	Korea	20	retrospective multicenter	235	PCNSL	63	records review	21 mo.	33	11.7% at 1 yr
Lekovic	2010	Serbia	10	retrospective single center	42	PMBCL	34	records review	47 mo.	15	35.7% at 6 mo.
Borchmann	2019	Germany	9	clinical trials	573	HL	36	trial data	12 mo.	173	3.3%
Gebhart	2014	Austria	21	retrospective multicenter	70	SMZL	n.a.	records review	n.a.	9	13%
Hultcrantz	2014	Sweden	29	population-based databases	2190	WM/LPL	74	ICD codes	10 yrs	92	2.1% at 1 y
Gangaraju	2019	USA	26	registry	734	NHL	49	patient questionaire	8.1 yrs	58	8.1% at 10 yrs
Yamshon	2018	International	24	meta-analysis	1433	NHL	66	published studies	n.a.	77	4.5% at 6 mo.
Zhang	2016	China	11	retrospective single center	565	lymphoma	n.a.	not specified	n.a.	40	7.1% PICC-related

Abbreviations: No., number: pts, patients; VTE, venous thromboembolism; mo., months; yrs, years; DBCL, diffuse large B cell lymphoma; FL, follicular lymphoma; NHL, Non-Hoddgkin lymphoma,; HL Hodgkin lymphoma; CLL, chronic lymphocytic leukemia; SMZL, splenic marginal zone lymphoma; PCNSL, primary central nervous system lymphoma; PMBCL, primary mediastinal B cell lymphoma; WM, Waldenstroems macroglobulinemia; LPL, lymphoplasmocytic lymphoma; ICD, international classification of diseases; IR, incidence rate; CI cumulative incidence; PICC, peripherally-inserted central catheter.

**Table 2 cancers-12-01291-t002:** VTE risk factors in lymphomas.

First Author	Ref. No.	Histology	Age	Gender	BMI	PriorVTE	Cmorb.	Stage	ECOG	LDH	Hb	WBC	Plt	KS	Other
Sanfilippo	25	DLBCL	No	n.d.	Yes	Yes	No	stage	n.d.	No	Yes	n.d	n.d.	No	
Santi	22	DLBCL	Yes	Female			n.d.	No	n.d.	n.d.	n.d.	n.d.	n.d.	Yes	
Antic	7	aggressive	No	No	Yes	Yes	n.d.	E, Med	Yes	n.d.	Yes	Yes *	No	n.d.	
Rupa-Matysek	12	DLBCL	No	No	n.d.	Yes	n.d.	Med	Yes	n.d.	Yes	Yes	No	n.d.	
Rupa-Matysek	13	DLBCL	No	No	n.d.	n.d.	n.d.	Bulk	n.d.	n.d.	Yes	Yes	No	No	IPI score
Hohaus	8	aggressive	Yes	No	n.d.	n.d.	n.d.	bulk, CNS	Yes	Yes	No	No	No	No	albumin < 4
Park	30	aggressive	Yes	No	n.d.	No	Yes	stage, E, Med, CNS	Yes	Yes	n.d.	n.d.	n.d.	n.d.	
Mohren	14	aggressive	No	No	n.d.	n.d.	n.d.	No	n.d.	n.d.	n.d.	n.d.	n.d.	n.d.	
Zhou	15	n.d.	No	Female	No	n.d.	No	No	n.d.	No	Yes	No	No	n.d.	creatinine
Mahajan	27	aggressive	Yes	No	n.d.	n.d.	Yes	stage	n.d.	n.d.	n.d.	n.d.	n.d.	n.d.	Asian
Lund	28	aggressive	No	No	n.d.	n.d.	No	CNS	Yes	Yes	No	No	No	n.d.	
Caruso	23	aggressive	n.d.	n.d.	n.d.	n.d.	n.d.	No	n.d.	n.d.	n.d.	n.d.	n.d.	n.d.	
Lim	6	n.a.	Yes	No	No	n.d	n.d.	stage, E	Yes	No	No	Yes	No	No	IPI score
Komrokji	16	n.a.	No	No	n.d.	n.d	n.d.	stage	No	No	n.d.	n.d.	n.d.	n.d.	IPI score
Borg	17	n.a.	n.d.	No	No	Yes	n.d.	stage	Yes	No	No	No	No	n.d.	IPI score
Yokoyama	19	n.a.	Yes	No	No	n.a.	n.d.	No	Yes	Yes	No	No	No	n.d.	IPI score
Goldschmidt	18	n.a.	No	n.d.	n.d.	n.a	n.d.	n.d.	n.d.	n.d.	n.d.	No	n.d.	n.d.	
Byun	20	n.a.	Yes	Female	No	n.a.	n.d.	n.d.	Yes	No	Yes	No	No	n.d.	albumin <4
Lekovic	10	n.a.	No	No	n.d.	n.a	n.d.	n.d.	n.d.	n.d.	n.d.	No	No	n.d.	fibrinogen
Borchmann	9	n.a.	n.d.	No	No	n.d	n.d.	Yes	n.d.	n.d.	No	No	No	No	
Gebhart	21	n.a.	n.d.	n.d.	n.d.	n.d	n.d.	n.d.	n.d.	n.d.	n.d.	n.d.	n.d.	n.d.	LAC
Hultcrantz	29	n.a.	n.d.	n.d.	n.d.	n.d	n.d.	n.d.	n.d.	n.d.	n.d.	n.d.	n.d.	n.d.	
Gangaraju	26	n.d.	n.d.	n.d.	Yes	n.d	Yes	n.d.	n.d.	n.d.	n.d.	n.d	n.d.	n.d.	GVHD

Abbreviations: BMI, body mass index; VTE, venous thromboembolism, Cmb, Comorbidities; ECOG, performance status according to ECOG scale; LDH, lactate dehydrogenase; Hb, hemoglobin; WBC, white blood cell count; plt, platelet count; KS, Khorana score; DLBCL, diffuse large B cell lymphoma; E, extranodal disease; Med, mediastinal involvement; IPI, international prognostic index; LAC, lupus anticoagulant; GVHD, graft versus host disease; n.d, not done; n.a., not assessable. * indicates that WBC < 1000/mm^3^ during chemotherapy was a risk factor instead of the WBC > 11,000/mm^3^ at diagnosis in the other studies. The studies by Yamshon and Zhang were not included into Table 2, as they do not report on VTE risk factors, but focus on particular clinical situations (therapy with lenalidomide and CVC).

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
