# Peer review of "Venous Thromboembolism in Lymphoma: Risk Stratification and Antithrombotic Prophylaxis"

_cancers, 2020, doi:10.3390/cancers12051291_

Round 1
Reviewer 1 Report
The present manuscript aims to review the available current data on the incidence and risk factors of VTE in lymphoma patients. The review is very detailed and well presented.
Based on the presented data, VTE is a significant complcation in lymphoma patients which requires further investigation and awareness by the clinicians. The present manuscript contributes to the knowledge of the incidence and risk factors of VTE as well as to the better management of this complicaion.
Author Response
We thank the reviewer for the evaluation.
No suggestions for modification had been made
Reviewer 2 Report
I reviewed the manuscript by Hohaus and collagues about VTE risk factors and prophylaxis in lymphoma. Some issues should be adressed before I can recommend this paper for publication in Cancers:
1.) Title: Given the increasing interest in arterial thrombosis in cancer patients, "thrombosis" alone may be misleading: I suggest the authors to use "venous thromboembolism" or "venous thrombosis" in the title, to eliminate any ambiguity and show readers that this is a review about venous thrombosis.
2.) Methods: It is somewhat odd that the authors perform a literature search without reporting details about the search methodology. In this way the current paper is neither a narrative review nor a systematic review, but a "hybrid" in between...I suggest that the authors at least provide a small supplementary appendix about their search strategy, including the used MESH terms etc.
3.) Tables 1 and 2 are full of bizarre symbols, obviously a conversion error from word to pdf. Please rectify.
4.) It is somewhat odd that the authors use so much text for describing risk factors, while the section on prophylaxis and treatment is not event one page long...several "hot topics" on treatment and prophylaxis are not discussed...I would add the following to section 6:
- What is the status of DOACs for therapy of cancer-associated VTE in lymphoma? If there are insufficient data on this, what is the authors' opinion on this issue?
- How does thrombopenia (often encountered in lymphoma patients during chemotherapy and VTE) influence the dosing of LMWH for treatment of cancer-associated VTE in lymphoma?
- What is the status of DOACs for prophylaxis of cancer-associated VTE in lymphoma ambulatory outpatients? If there are insufficient data on this, what is the authors' opinion on this issue?
- What about thromboprophylaxis for hospitalized lymphoma patients?
Author Response
We thank the reviewer for the evaluation of the manuscript and addressed all suggestions of and accordingly modified the manuscript.
1.) Title: Given the increasing interest in arterial thrombosis in cancer patients, "thrombosis" alone may be misleading: I suggest the authors to use "venous thromboembolism" or "venous thrombosis" in the title, to eliminate any ambiguity and show readers that this is a review about venous thrombosis.
We modified the title to “Venous Thromboembolism in lymphoma: risk stratification and antithrombotic prophylaxis”, as suggested.
2.) Methods: It is somewhat odd that the authors perform a literature search without reporting details about the search methodology. In this way the current paper is neither a narrative review nor a systematic review, but a "hybrid" in between...I suggest that the authors at least provide a small supplementary appendix about their search strategy, including the used MESH terms etc.
We agree that the review is a kind of mix: As there are few data on prophylaxis and treatment of VTE that are lymphoma-specific, this part of the review is narrative. As there is a growing body of literature on VTE risk factors in lymphoma, we performed a literature search to approach this in a more systematic way. We therefore deleted the term “systematic”.
The description of the search strategy was brief. As suggested we add a flow chart as figure 1 detailing the following information:
“To review VTE risk factors in patients with lymphoma, we screened the Pubmed database for reports published between Jan 1, 2000 to Dec 31, 2019, using the MeSH terms “lymphoma” and “thromboembolism” and “venous”. We reviewed 246 references. Publications addressing VTE risk in cohorts of adult patients with lymphoma were included for this review, and 21 studies were eligible Figure 1). Four additional studies were identified by cross-referencing in the 21 published reports. The articles were divided among all the authors for a first classification and summary and then reviewed by the two senior authors (S.H. and V.D.S.).
The flow chart in Figure 1 illustrates that 51 articles were excluded on the basis of the title. A further 174 articles were excluded, as they were case reports (n=28), studies on pediatric populations (n=27), studies on other malignancies than lymphoma (n=88), reports describing laboratory abnormalities without risk estimation (n=12), narrative reviews (n=15), and articles in languages other than English (n=3). After the screening, 21 studies were eligible. Four additional studies were identified by cross-referencing in the 21 published reports.
3.) Tables 1 and 2 are full of bizarre symbols, obviously a conversion error from word to pdf. Please rectify.
Tables were edited.
4.) It is somewhat odd that the authors use so much text for describing risk factors, while the section on prophylaxis and treatment is not event one page long...several "hot topics" on treatment and prophylaxis are not discussed.
The imbalance between risk factors and prophylaxis/treatment is due to the imbalance in available literature on these topics specific for lymphoma patients. Paragraphs on treatment and prophylaxis have been extended as outlined below.
I would add the following to section 6: What is the status of DOACs for therapy of cancer-associated VTE in lymphoma? If there are insufficient data on this, what is the authors' opinion on this issue?
We now screened the published studies on DOAC for the presence of lymphoma patients. We identified 4 randomized clinical trials comparing the use of DOAC in cancer-associated VTE and 2 trials on primary prophylaxis of cancer-associated VTE. Study details and outcomes are briefly discussed, and number of patients with lymphoma are detailed. We added the following paragraphs to page 15/16 on treatment and to page 16/17 on prophylaxis:
“The use of direct oral anticoagulants (DOAC) is emerging as a safe and effective alternative to subcutaneous LMWH for the treatment of cancer-associated VTE. Four randomized clinical trials showed that DOAC were noninferior to LMWH for the treatment of cancer-associated VTE without an increased risk of major bleeding [54–57]. Only few patients with hematological malignancies were enrolled, and the proportion of lymphoma patients was lower than 5%.
In the Hokusai VTE cancer study, 40 of 1050 (3.8%) patients had lymphoma [54]. Primary outcome was the composite of recurrent VTE or major bleeding. The DOAC edoxaban was not inferior to LWMH. In a recent subgroup analysis, data for 111 patients with hematological malignancies including the 40 patients with lymphoma were presented, and no difference in the primary outcome during the 12-month observation period between edoxaban and dalteparin was observed (8.9 and 10.9%) [58].
The SELECT D trial randomized 406 patients with active cancer and VTE to either rivaroxaban or dalteparin. [55]. No differences were observed between in the primary outcome that was recurrent VTE. Only 10 (2.5%) patients had hematological cancers.
In the ADAM VTE trial, the safety of the DOAC apixaban was compared to LMWH in 300 cancer-associated VTE [56]. Primary outcome was major bleeding, and secondary outcomes included VTE recurrence and a composite of major plus clinically relevant non-major bleeding (CRNMB). No major bleedings were observed in the apixaban arm, and VTE recurrence was lower in the apixaban arm. Only 16/300 (5.3%) patients had lymphoma.
In the Caravaggio trial apixaban was compared to dalteparin for the treatment of cancer-associated VTE [57]. Dalteaparin was noninferior to dalteparin without an increased risk of major bleeding. Of the 1155 enrolled patients, 85 (7.4%) had hematological malignancies, a subanalysis on the small number of hematological malignancies did not reveal a difference between the two arms.”
“Recently, DOACs have also been compared to placebo as primary prophylaxis to prevent cancer-associated VTE in high-risk patients, identified by the Khorana score (>2). In the AVERT trial, 574 patients were randomized, and the occurrence of VTE was lower in the apixaban group (4.2% versus 10.2%) while major bleeding was increased (3.5% versus 1.8%) [59]. A total of 145 (25.2%) patients with lymphoma were included. Outcomes according to disease groups were not reported. In the CASSINI trial randomizing 841 patients, rivaroxaban reduced the occurrence of VTE during the treatment period when compared to placebo (2.6% versus 6.4% ) without increasing the risk of major bleeding [60]. However, there was no difference in the VTE incidence when the analysis was extended to the full observation period of 180 days. Separate outcome for the 59 (7.0%) patients with lymphoma were not reported.”
How does thrombopenia (often encountered in lymphoma patients during chemotherapy and VTE) influence the dosing of LMWH for treatment of cancer-associated VTE in lymphoma?
We do not think that there are specific modifications for patients with lymphoma that differ from recommendations for patients with hematological malignancies and thrombocytopenia (Napolitano et al, 2019). In the individual bleeding/VTE risk evaluation, it is probably part of good clinical practice to consider risk factors potentially increasing the bleeding risk as gastrointestinal localization or potentially increasing the severity of bleeding as lymphoma localization to the CNS. However these considerations do not reach the evidence for a formal recommendation. We added the following paragraph to page 15
“Thrombocytopenia can develop during chemotherapy and treatment of VTE. An expert panel endorsed by the Gruppo Italiano Malattie Ematologiche dell'Adulto Working Party on Thrombosis and Haemostasis produced a formal consensus about platelet cut-offs for safe treatment with LMWH in thrombocytopenic patients [53]. Dose modifications of LMWH for platelet counts < 50x109/L are recommended. In clinical practice, risk factors for bleeding and VTE have be considered to balance the risk in the individual patient, as localization to particular sites could potentially increase the bleeding risk or the sequelae of bleeding as localization of the lymphoma to the gastrointestinal or the central nervous system.”
What is the status of DOACs for prophylaxis of cancer-associated VTE in lymphoma ambulatory outpatients? If there are insufficient data on this, what is the authors' opinion on this issue?
We think primary VTE prophylaxis with DOAC it is an interesting opportunity that has to be explored in randomized studies. Data are insufficient. Interactions with chemotherapeutic drugs drugs to metabolism via cytochrome P450 system is of concern. A prerequisite for the design of a randomized study is the validation of recently published VTE risk models for lymphomas, as the Khorana score for patients with solid tumors is less predictive. We added these considerations to page 17:
“Primary VTE prophylaxis with DOACs is an interesting perspective that has to be further explored in randomized studies specifically designed for patients with lymphoma. Potential interactions with chemotherapeutic drugs involving metabolism via cytochrome P450 system is of concern. A prerequisite for the design of a randomized study is the validation of recently published VTE risk models for lymphomas, as performance of the Khorana score is low in this category of patients. “
What about thromboprophylaxis for hospitalized lymphoma patients?
As we show in table 2, reduced mobility expressed by the ECOG performance status is a risk factor in 9/10 studies addressing this parameter. Therefore, we think hospitalization associated with conditions limiting the patient’s mobility should be a reason for thromboprophylaxis. We added this consideration on page 16:
“Reduced mobility expressed by the ECOG performance status is a risk factor in 9/10 studies addressing this parameter in patients with lymphoma (Table 2). Therefore, hospitalization associated with conditions limiting the patient’s mobility should be a reason for thromboprophylaxis.”
Reviewer 3 Report
Dear editor, thank you for the opportunity to review the systematic review presented of VTE and lymphoma by Hohaus and team. The manuscript is generally well-written and the topic is of interest, especially to hematologists. The authors present the incidence of VTE in this population and consequently discuss risk factors and risk models.
The article is nicely structured and the sentences read well. The manuscript is however a little boring as it misses real opinion and discussion. Recently, The Khorana score was used in two large RCTs evaluating thromboprophylaxis in cancer patients, including some lymphoma patients. Consequently the score is endorsed by guidelines for use in clinical practice. There is quite some discussion about the performance of this score. Is it working differently in solid and hematologic cancers? The score includes lymphoma as variable, should we now use this strategy in Lymphoma? Why of why not? Should we use the throly score? What is needed to start using scores in the lymphoma population? Is your own score performing better than the other scores?
Please emphasize the goal of this review. Can reader use this manuscript to improve prediction by using the risk scores?
Some minor points:
Methodology
In general, numbers of 10 and lower should be written as ‘ten’.
Since the authors claim to have performed a systematic review, I suggest to add the full search strategy (where MeSH terms used?) and flow diagram. Please see PRISMA checklist.
In Table 1 incidence rate is expressed as %, which is weird as incidence rates are often not expressed as percentages but as count per 1000 patient years/cycles. Do the authors mean cumulative incidence?
Epidemiology
When discussing VTE incidences, please add a time-windows of the estimation (throughout the whole manuscript) as this is crucial when interpreting numbers. Did patients in the studies receive thromboprophylaxis? Please clarify.
‘form China’ should read ‘for China’
Risk assessment models
The Khorana score includes lymphoma as variable, and is endorsed by various international guidelines. Please discuss this a little more. How well does the score perform in the non-lymphoma population? How well in the lymphoma population? Should it be used at this point?
I also miss the risk model of Pabinger et al (lancet haematology), which uses D-dimer for VTE prediction in a cohort which also comprised lymphoma patients.
The authors mention a risk score developed by themselves but do not comment on the predictive performance. Is it ever validated? Should it be used?
Author Response
We thank the reviewer for the evaluation of our manuscript. We addressed all the point raised by the reviewer and modifed the manuscript.
The article is nicely structured and the sentences read well. The manuscript is however a little boring as it misses real opinion and discussion. Recently, The Khorana score was used in two large RCTs evaluating thromboprophylaxis in cancer patients, including some lymphoma patients. Consequently the score is endorsed by guidelines for use in clinical practice. There is quite some discussion about the performance of this score. Is it working differently in solid and hematologic cancers? The score includes lymphoma as variable, should we now use this strategy in Lymphoma? Why of why not? Should we use the throly score? What is needed to start using scores in the lymphoma population? Is your own score performing better than the other scores?
Large RCT on thromboprophylaxis included too few patients to allow for a sensitivity analysis in this subgroup of patients. Moreover, the VTE risk in the lymphoma patients is not separately reported. We think that in the future we will have lymphoma-specifc scores that can be used, however further validation of ThroLy score or the score we developed is clearly needed before we can do so. In the meanwhile, it is still reasonable to use the Khorana score as inclusion criterion for studies on primary prophylaxis including a variety of malignancies.
Please emphasize the goal of this review. Can reader use this manuscript to improve prediction by using the risk scores?
The goal of this review is to illustrate the peculiar VTE risk factors in patients with lymphoma, in order to raise the awareness of the reader that VTE scores for patients with solid tumors might not perfectly fit the VTE risk situation in patients with lymphoma. As physicians increasingly specialize in the treatment of a few or single cancer type, knowledge on disease-specific risk factors will become more and more important to help treating physicians in their decision on VTE prophylaxis. We think that the reader could improve his clinical practice considering both single VTE risk factors, as the ThroLy score and the score we developed. Moreover, we hope that this review will stimulate the research community to assess, validate and improve risk scores in patients with lymphoma.
We added these considerations to page 3:
“VTE risk factors in lymphoma differ from VTE risk factors in solid tumors that have been used to build pan-cancer VTE risk scores, that do not capture the disease-specific VTE risk in lymphomas. As physicians increasingly specialize in the treatment of a few or single cancer type, knowledge on disease-specific risk factors will become more and more important to help treating physicians in their decision on VTE prophylaxis. More research is needed to assess, validate and improve VTE risk scores in patients with lymphoma.”
Some minor points:
Methodology
In general, numbers of 10 and lower should be written as ‘ten’.
Numbers were changed when indicated.
Since the authors claim to have performed a systematic review, I suggest to add the full search strategy (where MeSH terms used?) and flow diagram. Please see PRISMA checklist.
We now include a more detailed description of our literature research and a flow chart as appendix as described above.
In Table 1 incidence rate is expressed as %, which is weird as incidence rates are often not expressed as percentages but as count per 1000 patient years/cycles. Do the authors mean cumulative incidence?
Table 1 shows cumulative incidences, and consequently, we changed the term as suggested.
Epidemiology
When discussing VTE incidences, please add a time-windows of the estimation (throughout the whole manuscript) as this is crucial when interpreting numbers. Did patients in the studies receive thromboprophylaxis? Please clarify.
We added the time windows for cumulative incidences to Table 1. Generally, about 95% of VTE are observed at diagnosis or during treatment that lasts about 6 months, and therefore cumulative incidences that are given either at 6 months, 1 year, 2 years or during therapy should be comparable.
We agree that data on thromboprophylaxis would be of great interest, however studies presented in table 1 do not report data on thromboprophylaxis with exception of Zhou et al reporting a 4.3% of patients receiving thromboprophylaxis. Therefore, one cannot appreciate a potential effect of thromboprophylaxis on VTE risk. We added this consideration to page 5:
“Thromboprophylaxis with low molecular heparin (LMWH) might modify VTE risk, however the studies on VTE risk factors do not report sufficient data to evaluate the role of thromboprophylaxis”
‘form China’ should read ‘for China’
“Form China ” was changed to “from China”
Risk assessment models
The Khorana score includes lymphoma as variable, and is endorsed by various international guidelines. Please discuss this a little more. How well does the score perform in the non-lymphoma population? How well in the lymphoma population? Should it be used at this point?
As a complete discussion of the Khorana score in non-lymphoma patients is beyond the scope of the review, we will briefly discuss the role on performance of the Khorana score in non-lymphoma patients, as it may be of interest to the lymphoma population and extend our discussion on the potential use of risk models in lymphoma patients, as follows to pages 15 and 16:
“In the same line, analyses of patient cohorts focusing on specific cancer sites, as pancreatic cancer or lung cancer, showed a poor performance of the Khorana score [46,47]. A recent meta-analysis revealed that the predictive power of the Khorana score was not homogenous across various types of cancer [48]. The pan-cancer Khorana score does not capture the disease-specific characteristics associated with VTE risk, and clinicians should be cautious when applying the Khorana score as a universal risk assessment tool.
Moreover, the sensitivity of the Khorana score is quite poor. Most VTE events in cancer patients occur outside the high-risk group [49]. In a meta-analysis on 27,849 patients, only 23.4% of VTE occurred in the 17.4% of patients with a high-risk (>3) Khorana score [49]. When decreasing the threshold to include also patients with a risk score of 2, the detection rate increased to 55.2% of VTE events in a patient cohort that included 47% of the cancer population [49].”
“Similar to the Khorana score, the performance of the ThroLy score is limited by a high frequency of VTE occurring in the low-risk group. In a validation study by Rupa-Matysek, 48% of VTE occurred in the low-risk group of the ThroLy score that comprised 75% of patients [12]. “
“The predictive performance of our score is quite good when compared to the Khorana and ThroLy score. Our VTE score identified 82% of VTE in the high-risk group that consisted of 48% of patients [8]. If the performance is confirmed in validation studies, this score will be a simple tool for VTE risk stratification and could be useful in designing studies on primary prophylaxis for higher risk patients.”
I also miss the risk model of Pabinger et al (lancet haematology), which uses D-dimer for VTE prediction in a cohort which also comprised lymphoma patients.
We thank the reviewer for this suggestion and now include a discussion on the VTE risk model developed by Pabinger et al.: We added the following paragraph to page 14:
“Pabinger et al identified tumor-site risk category as the only parameter of the Khorana score to predict VTE in the Cancer and Thrombosis Study (CATS) cohort of 1423 patients [52]. Combining tumor-site category and D-dimer values the authors developed a simple and VTE risk model validated in an independent patient cohort. The CATS cohort included 249 patients with lymphoma (17%) while the validation cohort of 832 patients did not contain lymphoma patients. This VTE risk score merits further exploration in lymphoma patents.”
The authors mention a risk score developed by themselves but do not comment on the predictive performance. Is it ever validated? Should it be used?
As above outlined, the predictive performance of our score is quite good when compared to the Khorana and ThroLy score. As it has bene published in 2019, no validation study is available so far. Once validated, we think it will be a simple tool for VTE risk stratification. We added the following consoideration on page 14:
“The predictive performance of our score is quite good when compared to the Khorana and ThroLy score. Our VTE score identified 82% of VTE in the high-risk group that consisted of 48% of patients [8]. If the performance is confirmed in validation studies, this score will be a simple tool for VTE risk stratification and could be useful in designing studies on primary prophylaxis for higher risk patients.”
Round 2
Reviewer 2 Report
The authors have thoroughly adressed my suggestions, I can recommend the manuscript now for publication in "Cancers."
Reviewer 3 Report
The authors improved the manuscript substantially. I have no further comments.